# How Attitudes toward Alcohol Policies Differ across European Countries: Evidence from the Standardized European Alcohol Survey (SEAS)

**DOI:** 10.3390/ijerph16224461

**Published:** 2019-11-13

**Authors:** Carolin Kilian, Jakob Manthey, Jacek Moskalewicz, Janusz Sieroslawski, Jürgen Rehm

**Affiliations:** 1Institute of Clinical Psychology and Psychotherapy, TU Dresden, 01187 Dresden, Germanyjtrehm@gmail.com (J.R.); 2Centre for Interdisciplinary Addiction Research, UKE Hamburg-Eppendorf, 20246 Hamburg, Germany; 3Institute of Psychiatry and Neurology, 02-957 Warsaw, Poland; sierosla@ipin.edu.pl (J.S.); moskalew@ipin.edu.pl (J.M.); 4Institute for Mental Health Policy Research, Centre for Addiction and Mental Health, Toronto, ON M6J 1H4, Canada; 5WHO Collaboration Centre, Centre for Addiction and Mental Health, Toronto, ON M5S 2S1, Canada; 6Institute of Medical Science, University of Toronto, Toronto, ON M5S 1A8, Canada; 7Campbell Family Mental Health Research Institute, Centre for Addiction and Mental Health, Toronto, ON M5G 2C1, Canada; 8Department of Psychiatry, University of Toronto, Toronto, ON M5T 1R8, Canada; 9I.M. Sechenov First Moscow State Medical University (Sechenov University), Alexander Solzhenitsyn Street 28/1, 109004 Moscow, Russian

**Keywords:** alcohol policy, alcohol policy endorsement, attitudes toward alcohol control policy, Europe, latent class analysis

## Abstract

Alcohol policy endorsements have changed over time, probably interacting with the implementation and effectiveness of alcohol policy measures. The Standardized European Alcohol Survey (SEAS) evaluated public opinion toward alcohol policies in 20 European locations (19 countries and one subnational region) in 2015 and 2016 (*n* = 32,641; 18–64 years). On the basis of the SEAS report, we investigated regional differences and individual characteristics related to categories of alcohol policy endorsement. Latent class analysis was used to replicate cluster structure from the SEAS report and to examine individual probabilities of endorsement. Hierarchical quasi-binomial regression models were run to analyze the relative importance of variables of interest (supranational region, gender, age, educational achievement, and drinking status) on class endorsement probability, with random intercepts for each location. The highest support for alcohol control policies was recorded in Northern countries, which was in contrast to the Eastern countries, where the lowest support for control policies was found. Across all locations, positive attitudes toward control policies were associated with the female gender, older age, and abstaining from alcohol. Our findings underline the need to communicate alcohol-related harm and the implications of alcohol control policies to the public in order to increase awareness and support for such policies in the long run.

## 1. Introduction

In Europe, alcohol is one of the major risk factors for disease burden, although, in general, alcohol consumption, as well as irregular heavy drinking, have decreased in the last years [1,2], Europe is still the region with the highest consumption levels worldwide [2]. In order to reduce alcohol consumption further, the “European action plan to reduce the harmful use of alcohol 2012–2020” [3] lists 10 action points which aim to improve national alcohol policies in European countries, and therefore reduce risky drinking behaviors. The 10 action points involve enhancing the population’s awareness and commitment, improving health services’ responses, encouraging community and workplace actions, implementing drinking-driving policies and countermeasures, reducing the availability of alcohol, instituting marketing restrictions, increasing pricing, reducing the negative consequences of drinking and alcohol intoxication and the public health impact of illicit alcohol, and, finally, improving monitoring and surveillance efforts. In 2016, there was a relatively high overall implementation of alcohol policies, however, some of the most effective and cost-effective policies such as pricing or marketing regulations [4] lagged behind [1,5].

The effectiveness of key alcohol policy measures has been substantiated by a large body of reviews and meta-analyses, although they vary in specific aims (e.g., reducing alcohol intake directly or indirectly by enhancing awareness of alcohol-related harm, see [6,7]). However, public opinions, such as perceived effectiveness of and attitudes toward alcohol policies, which probably interact with the implementation and effectiveness of some alcohol policy measures [8,9,10], are sometimes neglected by researchers but play an important role for policymakers. In fact, alcohol policy endorsement is not static but changes over time [11,12,13,14] and routine monitoring of public opinion regarding alcohol policies can contribute to successful implementation of alcohol policy measures. As discussed by Giesbrecht and Livingston, a shift in public opinion can support changes in alcohol policies, and conversely changes in alcohol policies can also alter attitudes toward alcohol policies [8]. The public acceptance of policy measures depends on the respondents’ knowledge of the expected outcomes, as demonstrated in the case of minimum unit pricing (MUP) for alcohol in the UK [15]. The acceptability of MUP have been increased significantly when the expected outcomes of the policy are clearly outlined for the respondents. In addition, beliefs regarding the effectiveness of alcohol policy measures and beliefs regarding the harm caused by drinking have been demonstrated to affect attitudes toward alcohol policy measures [16,17].

To date, several publications have focused on sociodemographic and demographic differences of attitudes toward alcohol policies and on the relationship between drinking behaviors and alcohol policy endorsement [16,17,18]. In general, based on these studies, women and older adults were more likely than men and younger adults to support alcohol policies [16]. Furthermore, individuals with higher educational levels have reported higher rates of endorsement of restrictions on alcohol use as compared to individuals with lower educational levels [17]. Heavier drinkers showed higher rates of rejection of restrictions and taxation as compared with individuals who reported light drinking or abstaining [16,18]. However, previous findings are often limited to a low number of countries, meaning that country-specific factors, such as national alcohol policies, could not be taken into account.

In this study, we used data on attitudes toward alcohol policies from 19 European countries and one subnational region to examine different clusters of alcohol policy endorsement, to compare them between supranational regions, and to evaluate their relationship to individual respondent characteristics. The data were collected during the course of the Standardized European Alcohol Survey (SEAS), which was implemented as part of the EU Joint Action on Alcohol known as the “Reducing Alcohol Related Harm” (RARHA) project in 2015 and 2016. Significant variations among attitudes toward alcohol policies across the 20 European locations were demonstrated in the published Synthesis report, which identified the following three clusters of attitudes through factor analyses using the same data (Table 1): Population-based alcohol control policies (e.g., support for controlling the number of places selling alcohol), education and individual-based alcohol policies (e.g., support for printed warnings about alcohol-related harms), and laissez-faire alcohol policies (e.g., support for individual responsibility) [19].

Using the RARHA SEAS data, this study pursued two aims which were: (1) to compare clusters of alcohol policy measures between different European regions and (2) to analyse associations between categories of alcohol policy endorsement and individual characteristics (gender, age, educational achievement, and drinking status) taking into consideration national alcohol policy scores. Building on previously identified factor structure of policy endorsement [19], i.e., population-based alcohol control, education and individual-based alcohol policies, and laissez-faire alcohol policies, we expected to identify the same three clusters in the latent class analyses (LCAs). LCA is a probabilistic model-based clustering approach which identifies individual class membership probabilities for each individual belonging to each class instead of single class membership as defined in factor analysis [20]. We were, therefore, were able to investigate associations between (sociodemographic and demographic characteristics and clusters of alcohol policy endorsement in all respondents using class membership probabilities. At the individual-level, we investigated the following hypotheses based on past publications: (a) Women and (b) older adults are more likely to endorse classes, which are most related to the factors “population-based alcohol control policies” and “education and individual-based alcohol policies”, whereas men and respondents of younger age are more likely to favor “laissez-faire alcohol policies” and (c) Abstainer and (d) respondents with a higher educational achievement are more likely to endorse “population-based alcohol control policies”’ in contrast to drinker and respondents with lower educational levels.

## 2. Materials and Methods

### 2.1. Data Collection

Survey data were used from the RARHA SEAS in 2015 and 2016, where 33,237 adults from 19 European countries and one region (Autonomous Community of Catalonia) participated (see Table 2 for sample characteristics by location).

In all locations, randomized sampling procedures were applied to select a representative general population sample aged 18 to 64 years. The sample size was about 1500 people in most locations, with few exceptions. The mode of administration differed between locations, but for most locations either computer-assisted telephone interviews or computer-assisted personal interviews were applied.

To assess individual consumption levels for the past 12 months, respondents provided data on their usual intake of three basic beverage types (beer, wine, and spirits using the beverage-specific quantity frequency approach). In addition, questions on frequency of risky single-occasion drinking (RSOD) were asked assuming a RSOD threshold of 40 grams of pure alcohol for women and 60 grams for men. Both measures were combined to estimate annual consumption of pure alcohol. To avoid overestimation of individual consumption, capping procedures were applied at 0.5 L of pure alcohol for daily alcohol intake of any alcoholic beverage, and at 182.5 L of pure alcohol for annual consumption.

In our analyses, we considered the following covariates: Gender (women and men), three age groups (≤34 years, 35 to 49 years, and ≥50 years), socioeconomic status as defined by educational achievement (primary and lower secondary education, secondary education, and high education), and drinking status grouped into (past-year) abstainers, low, and high-risk drinking (intake of pure alcohol per drink day, women ≥ 20 g and men ≥ 40 g). Individuals with missing data on gender, educational achievement, or quantity of drinking were excluded (*n* = 457, 1.4% of the sample).

### 2.2. Attitudes Toward Alcohol Policies

Attitudes toward eleven different alcohol policy measures were assessed (see Table 1 for details). Respondents identified how strongly they agreed or disagreed with each statement (strongly agree, somewhat agree, somewhat disagree, and strongly disagree). For statistical analyses, the variables were dichotomized and coded as “endorse”, if a respondent indicated “somewhat agree” or “strongly agree”, or as “not endorse”, if a person indicated “somewhat disagree” or “strongly disagree”. For the current contribution, we considered three clusters of attitudes toward alcohol policies based on the factor analysis reported in the Synthesis report [19]: Population-based alcohol control policies, education and individual-based alcohol policies, and laissez-faire alcohol policies. For a total number of 32,641 respondents (i.e., 0.42% missing values on the policy endorsement variables), data was available and analyzed.

### 2.3. Country-Level Variables

Country-specific alcohol policy scores were taken from the AMPHORA project [21]. The summary score represents the degree of implementation of six alcohol policy measures in 2010, namely, (1) control of production, retail sale, and distribution of alcoholic beverages; (2) age limits and personal control; (3) control of drunk driving; (4) control of advertising, marketing, and sponsorship of alcoholic beverages; (5) public policy; and (6) alcohol taxation and price, which were taken into account using different weights (for details, see [21]). The resulting alcohol policy score indicates three levels of strictness of alcohol policy (liberal, medium, and strict) whereby the strictest policies were identified in the Nordic countries (e.g., Finland) and the most liberal in Central (e.g., Germany) and South Europe (e.g., Italy). For the Autonomous Community of Catalonia, we used the same score as for Spain.

The assignment of countries to European regions was based on the AMPHORA classification as well. Iceland, Finland, Norway, and Sweden were grouped as Nordic countries. Denmark, which is usually classified into this group, was included in the West and Central European region because of the more liberal alcohol policies found there as compared with the other Nordic countries. Austria, France, and the UK were also associated with the West and Central European region. Greece, Italy, Portugal, Spain, and the Autonomous Community of Catalonia were included in the Southern European region and the remaining countries formed the Eastern European region (Bulgaria, Croatia, Estonia, Hungary, Lithuania, Poland, and Romania).

### 2.4. Statistical Analyses

LCAs were run to identify the cluster structure of policy endorsement. LCAs are defined by the prevalence of each class and the probability that an individual in a certain class would endorse a certain item [20]. Therefore, this is a person-centered probabilistic approach which is not exclusive to single-class membership (i.e., in contrast to exploratory factor analysis). Every class is described by all variables of interest based on different endorsement probabilities. In our investigation, responses to the eleven items assessing alcohol policy endorsement were used to identify latent classes. Because previous factor analysis suggested that all policy measures could be grouped into three distinct categories, a three-class LCA model was performed. Weights were taken into consideration to account for sampling bias. On the basis of the fitted LCA model, we obtained three variables ranging between 0 and 1, indicating the probability of each individual belonging to each class. Those probabilities were used as dependent variables in the following regression analyses.

Two sets of regression models were run. In the first set, three hierarchical regression models were calculated to compare European regions (independent variable) for the probabilities to endorse in each of the three classes (dependent variable). In the second set of hierarchical regression analyses, the same three dependent variables, gender, age, educational achievement and drinking status were included as predictors. The latter set of models were adjusted to the national alcohol policy score to control for effects of local policy strictness. As the dependent variable ranged between 0 and 1, we fitted a logistic regression with standard error estimation based fractional response regression models [22]. In these models the intercept was allowed to vary across locations (i.e., random intercept model). Survey weights were applied in all regression analyses. All statistical analyses were performed using Stata 15.1 [23] and R version 3.6.1 [24].

## 3. Results

### 3.1. Results of the Latent Class Analysis

The probabilities for each item belonging to a particular class are displayed in Figure 1. In the first class, items related to control strategies such as selling limitations and the responsibility of public authorities had high probabilities, whereas the probabilities of individual responsibility, responsibility of parents toward their children, and the evaluation of alcohol as a commodity as any other, were very low. Given this pattern, we linked the first class to the factor “population-based alcohol control policies”. With respect to the second and third class, assignment to the original factors reported in the Synthesis report was less clear. The second class contained low probabilities for control strategies including limitation of selling places and hours or taxation, which reflects a pattern of “laissez-faire strategies”, and at the same time high probabilities for the items “education” and “random breath-testing”, which originally were related to the “education and individual-based alcohol policy” factor. In the third class, medium to high probabilities were identified for all items, even if they were contradictory (e.g., “parents, and no legal authorities, should decide at what age their child is allowed to drink alcoholic beverages“ versus “public authorities have the responsibility to protect people from being harmed by their own drinking”), which did not correspond to any of the previously identified factors. Such differences in cluster structure resulting from latent class analysis versus factor analysis are not unusual as they are based on different methodological approaches [20]. Due to this discrepancy with the original naming of factors, we indicated the first class as “support of alcohol control policies”, the second class as “rejection of alcohol control policies”, and the third class as “acquiescence tendency”. The average prevalence of endorsement of alcohol control measures (class 1) was 32.0% for the whole sample. The second class was the most common class (46.7%), whereas the third class was the smallest in our sample, with a prevalence for endorsement of 21.4%. In spite of the fact that attitudes rejecting alcohol control measures were the most prevalent, a crucial statement in this area that “alcohol is a product as any other and does not require any special restrictions” has relatively low endorsement probabilities in all classes, not surpassing 50% in class two and three, and below 5% in class one.

### 3.2. Regional Differences in Alcohol Policy Endorsement

In Figure 2, the average probability of alcohol policy endorsement for each European region is presented by class. Across all regions, “rejection of alcohol control policies” was the most prevalent class, except for Nordic European countries. Conversely, support of alcohol control policies was considerably larger in Nordic European as compared with Eastern European (*OR* = 0.47, 95% *CI* [0.26; 0.85], *p* < 0.05) but not statistically different than West and Central European (*OR* = 0.62, 95% *CI* [0.32; 1.20], *n.s.*), or Southern European locations (*OR* = 0.58, 95% *CI* [0.31; 1.09], *n.s.*). Least supported was the third class (“acquiescence tendency”), which had the lowest proportion in all European regions. The proportion of endorsement in this latter class was significantly lower in the Nordic European region contrasting the Eastern (*OR* = 2.76, *p* <0.001, 95% *CI* (1.57 and 4.86), *p* < 0.001) and Southern European region (*OR* = 2.99, 95% *CI* (1.64 and 5.46), *p* < 0.001) but not compared to the West and Central European region (*OR* = 1.23, 95% *CI* (0.65 and 2.32), *n.s.*). No regional differences were observed with regard to the second class (Eastern European region, *OR* = 1.04, 95% *CI* (0.55 and 1.97), *n.s.*; West and Central European region, *OR* = 1.23, 95% *CI* (0.70 and 2.96), *n.s.*; Southern European region, *OR* = 0.85, 95% *CI* (0.43 and 1.68), *n.s.*; and Nordic European region, reference category).

### 3.3 Individual-Level Differences in Alcohol Policy Endorsement

Results of the hierarchical regression models linking respondents’ characteristics to the policy endorsement probability by class are presented in Table 3. Support for alcohol control policies was most likely to be given by women, middle-aged adults, respondents with higher educational levels, and abstainers. On the contrary, alcohol control policies were more likely to be rejected by men, younger adults, respondents with a higher education, and drinkers (i.e., low- and high-risk drinker) as compared with abstainers who are more likely to support this class representing a rejection of alcohol control policies. With regard to the third class (“acquiescence tendency”), higher odds for support were identified for women, middle-aged and older adults, respondents with a primary or lower secondary education, and abstainers.

## 4. Discussion

Our findings suggest that attitudes toward alcohol policy vary significantly between European regions. We identified three classes of alcohol policy endorsement which did not fully overlap with the previously identified factor structure. The three classes referred to support and rejection of alcohol control policies in addition to an “acquiescence tendency” class, which we discuss below. Overall, rejection of alcohol control policies was the most prevalent class, followed by its antipode, support for alcohol control policies. We showed that the support for alcohol control policies was mostly driven by Nordic European countries, while Eastern European countries showed significantly less support. In accordance with the dichotomy of both classes, two contrary patterns of those who support or reject control policies were identified as follows: Women, middle-aged adults, and abstainer preferably supported alcohol control policies, in contrast to men, adults aged younger than 35 years, and drinkers, who were more likely to reject alcohol control policies. Notably, educational achievement was not predictive of either support or rejection of alcohol control policies.

For a comparison of the current findings with the Synthesis report [19], methodological differences should be taken into account. In the current study, classes were characterized by the endorsement probabilities of all items of interest, instead of a selection as is usual in factor analysis. Therefore, the class structure and their properties differ from those presented in the Synthesis report. For example, belonging to the first class (i.e., support of alcohol control policies) means to report more support for items with high endorsement probabilities for this class (e.g., taxation, *p* = 69.9%) and less support for items with low endorsement probabilities (e.g., alcohol is a commodity as any other, *p* = 4.7%) in the survey. As a result of using a divergent approach, we were not able to replicate the cluster structure from the Synthesis report. Particularly critical is the third class where all items questioned had high endorsement probabilities. Since some items are contradictory to each other in this class (e.g., “parents, and no legal authorities, should decide at what age their child is allowed to drink alcoholic beverages“ versus “public authorities have the responsibility to protect people from being harmed by their own drinking”), we assumed that this class might reflect an acquiescence tendency among respondents. Bias due to acquiescence can occur for several reasons; the stimuli (e.g., cognitive load to the respondent), the respondent (e.g., education), or country-level indicators (e.g., collectivism) can be sources for higher acquiescence [25,26]. Remarkably, previous studies reported similar patterns of individual [25] and cross-country variations [27] in the acquiescence tendency in Europe as we found for class 3. For example, Van Herks and colleagues described a higher acquiescence tendency in the Mediterranean region than in the Northwestern European region. However, the high endorsement probabilities to all items in this class could also arise from the respondents’ tendency to at least “somewhat agree” with the policy statements, which would not be visible after a dichotomization of the response options.

Some further limitations have to be taken into consideration. First, the representativeness of survey data is limited due to the following reasons: (a) Samples are not representative of the entire population because they exclude certain groups of people, including the homeless and the imprisoned or otherwise institutionalized persons [28]; (b) alcohol surveys have a high nonresponse rates, often exceeding 50%, which was shown to be related to an underrepresentation of certain groups of people (i.e., individuals with low income or heavy episodic drinking [29]); and (c) there is an undercoverage of reported alcohol consumption compared to “real consumption” estimates [30,31]. Furthermore, limited generalizability of the results has to be considered due to great variations in the population size between locations, and therefore they might be not transferable to the whole European region. Finally, we investigated associations, and therefore we cannot draw conclusions about the direction of effects or why individuals endorse particular alcohol policies.

Our study presents evidence for variations in alcohol policy endorsement across European regions. In addition to alcohol policies being traditionally stricter in the Nordic region as compared to other European countries [21], we show that restrictive alcohol measures received more support in this region than elsewhere in Europe. In all studied regions, alcohol policies are considerably more liberal and control strategies were more likely to be rejected. Several publications reported a substantial increase in supporting alcohol control policies in Sweden, Finland, and Norway, since the millennium [12,13,14,32]. Although positive attitudes toward those policies increased, alcohol politics in Nordic countries were characterized by liberalization. But why did liberal alcohol policies lead to an increase in support for alcohol control measures there? A possible explanation is that individuals who experienced alcohol-related harm in their personal environment are more likely to endorse restrictive alcohol policies [33]. Referring to the Nordic countries, consumption levels and alcohol-related harm increased during the time period, where alcohol policies became more liberal [34,35,36]. With alcohol-related harm becoming more prevalent, more individuals are directly or indirectly affected by harm, which could lead to higher levels of awareness and, consequently, to alcohol control policy endorsement. In addition, even the knowledge about alcohol-related harm can predict changes in attitudes [17]. Another mediator discussed in the literature is a change in the individuals’ beliefs on the effectiveness of alcohol control strategies [11,16].

On the other side, our results indicate that in the Eastern European region there is considerably lower endorsement toward those restrictive policy measures although this is the region with the highest alcohol-related harm and alcohol-attributable mortality in Europe [1]. It can be argued, that the mediators explained above, i.e., the knowledge on alcohol-related harm and beliefs on the effectiveness of alcohol control strategies, are less widespread in the general population. This hypothesis goes hand in hand with a high prevalence of general acquiescence found in this region in our analyses, whereas the rejection of alcohol control policies was not significantly higher than in the other European regions. Another explanation can be related to general attitudes toward market economy and beliefs in self-regulatory powers of the market or support for liberal, laissez faire economic policies. In all countries which underwent transitions to market economy, the 1990s saw rapid liberalization of alcohol policies and dismantling of a previous control mechanism despite visible growth of alcohol-related problems [37]. In the decades of transitions toward market economy in Poland, the attitudes toward alcohol policy has changed dramatically. The general population survey carried out in 1992 indicated that only 12.8% of respondents endorsed a statement that “alcohol beverages should be treated as all other commodities and their sales should not be restricted” while 72.2% confirmed that “alcohol beverages must not be treated as all other commodities and their sales should be restricted by the State” [38]. Twenty-five years later in the RARHA SEAS survey, 62% of Polish respondents endorsed that “alcohol is a product like any other and does not require any special restrictions” [19]. The Polish experience, as well as experiences of other countries in transition, suggest that prevailing economic ideologies have a crucial impact on attitudes toward alcohol policy [39]. After decades of liberal policies, many Eastern European countries have considered returning to more restrictive policies. In a recent publication, increasing support of evidence-based alcohol control policies by the members of the Lithuanian Parliament were observed in the years between 2016 and 2018 [40]. In addition, with regard to the implementation of alcohol control policies in Lithuania between 2004 and 2019, traffic harm, injury, and mortality attributable to alcohol have been found to decrease ever since [41]. Nevertheless, studies on attitudes toward alcohol policies are scarce in most European countries. In light of our results, it would be interesting to study the interaction of public opinion and policy implementation and examine, for instance, if the former followed or influenced the latter.

We further investigated individual-level differences in alcohol policy endorsement. Our findings are in line with previous studies [16,17,18] and expand existing knowledge on individual characteristics associated with attitudes toward alcohol policies for a large number of European countries. Only one curious finding has to be discussed as both supporting (class 1) and rejecting alcohol control policies (class 2) were related to higher levels of education. We suggest that this is based on a higher tendency for acquiescence by individuals with a primary or lower secondary education [25]. This was underlined in the regression analysis, since the probability of endorsing the third class (i.e., acquiescence tendency) was up to twice as high in this subgroup as compared with higher educated respondents. However, in terms of individual differences such as gender, age, and educational achievement, it should be noted that they explain only a little about individual attitudes to alcohol policies as compared to individual differences in beliefs regarding alcohol-related harm and the effectiveness of alcohol-control measures [17]. Moreover, the framing of policies, i.e., how and who presented them to the public, is an important influencing factor to garner public acceptance and higher levels of alcohol policy endorsement [16]. These changes in public acceptance can be accomplished by targeting those who have higher levels of false beliefs and negative opinions to enhance alcohol policy endorsement more broadly.

## 5. Conclusions

In this study, we investigated regional differences and individual characteristics related to categories of alcohol policy endorsement in Europe. Our findings suggest that there are still considerable differences in endorsements toward alcohol control policies between European regions, with the highest level of support in the Nordic European region and lowest in the Eastern region. Since national alcohol policies across the EU are currently undergoing a change, transitions in the public opinion toward alcohol policies should be monitored to evaluate implementation processes. Future research should also focus on national and supranational factors that can influence public attitudes toward alcohol policies, such as a growing proportion of individuals with higher levels of education in high-income countries or changes in socioeconomic inequalities. To promote positive attitudes toward alcohol control policies in the public, future actions should include alcohol education about these evidence-based measures and their effectiveness on the harm which is attributable to alcohol, and, first of all, their positive impact on public health. They should be adapted to the status of implementing policies (i.e., before versus during the implementation process) and to the current national level of policy endorsement. A positive framing of alcohol policies before and during implementation is a key point of action.

## Figures and Tables

**Figure 1 ijerph-16-04461-f001:**
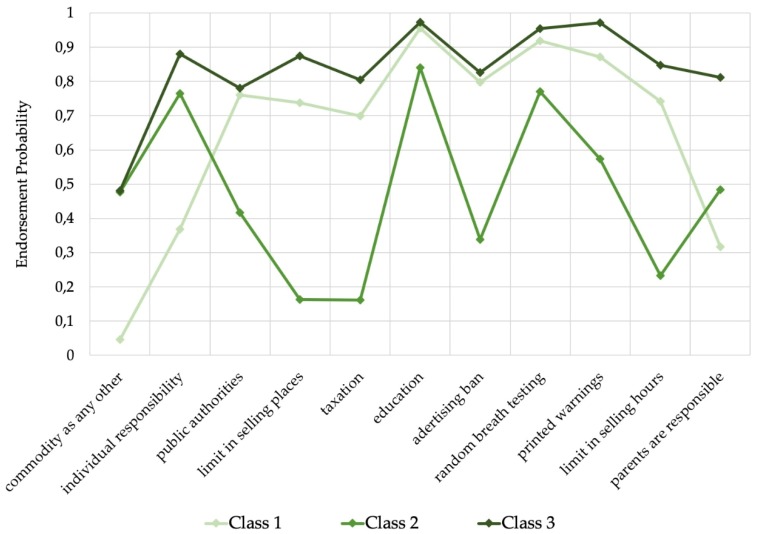
Results from the latent class analysis: Endorsement probability for each item in each class. Survey items are displayed on the x axis. Class 1 = ‘support of alcohol control policies’, class 2 = ‘rejection of alcohol control policies’, class 3 = ‘acquiescence tendency’.

**Figure 2 ijerph-16-04461-f002:**
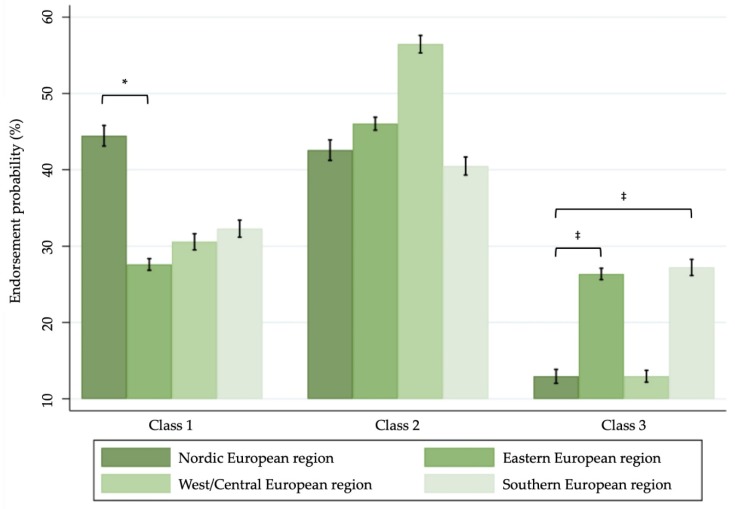
Average probability and confidence intervals to endorse the different classes of alcohol policies in the European regions by class. Class 1 = ‘support of alcohol control policies’, class 2 = ‘rejection of alcohol control policies’, class 3 = ‘acquiescence tendency’. Significance of regional differences based on hierarchical regression analyses for European region as predictor for class endorsement probability; Nordic European region was the reference category. * *p* < 0.05; ^†^
*p* < 0.01; ^‡^
*p* < 0.001.

**Table 1 ijerph-16-04461-t001:** Attitudes toward alcohol policies: Factor structure derived from the Synthesis report, “Comparative monitoring of alcohol epidemiology across the EU. Baseline assessment and suggestions for future action” ([19], p. 228) and corresponding items from the Standardized European Alcohol Survey.

Population-Based Alcohol Control Policies	Education and Individual-Based Alcohol Policies	Laissez-Faire Alcohol Policies
Public authorities have the responsibility to protect people from being harmed by their own drinking	Alcohol education and information should be the most important policy to reduce alcohol-related harm	Alcohol is a commodity as any other and does not require any special restrictions
The number of places selling alcohol should be kept low in order to reduce alcohol-related harm	Police should be allowed to check randomly if a driver is sober or not even without any indication of drunken driving	Adult people are responsible enough to protect themselves from harm caused by their drinking
Prices of alcoholic beverages should be kept high in order to reduce alcohol-related harm	Printed warnings about alcohol-related harm should be displayed on alcoholic beverages	Parents, and no legal authorities, should decide at what age their child is allowed to drink alcoholic beverages
Advertising of alcoholic beverages should be banned		
There should be limits on how late in the evening you can buy alcohol		

**Table 2 ijerph-16-04461-t002:** Sample size, response rate, demographic characteristics, educational achievement, and national alcohol policy score by locations in the RARHA SEAS.

Location	Sample Size ^a^	Response Rate (%)	Gender (% women)	Mean Age (*SD*)	Educational Achievement ^b^ (%)		National Alcohol Policy Score (ranking) ^c^
					Secondary Education	High Education	
Austria	3406	32.1	50.1	41.28 (0.23)	70.8	15.4	liberal
Bulgaria	3000	75.0	50.7	41.53 (0.24)	58.3	29.9	liberal
Croatia	1500	50.6	50.1	41.52 (0.36)	40.9	17.7	medium
Denmark	1575	52.5	51.9	40.93 (0.28)	48.7	37.5	medium
Estonia	2153	60.4	49.4	42.11 (0.38)	63.8	29.4	medium
Finland	1500	11.5	51.0	41.81 (0.36)	58.7	29.9	strict
France	1701	44.5	53.5	43.53 (0.35)	59.3	30.2	medium
Greece	1519	27.0	50.2	41.73 (0.34)	51.1	34.8	liberal
Hungary	2005	43.0	50.2	41.68 (0.33)	42.1	9.5	liberal
Iceland	873	47.7	49.4	40.23 (0.52)	38.4	45.5	strict
Italy	1468	8.7	50.3	42.52 (0.35)	55.8	17.7	liberal
Lithuania	1513	35.0	51.7	41.17 (0.36)	67.9	24.8	medium
Norway	1493	12.0	48.7	40.62 (0.34)	45.7	48.3	strict
Poland	1555	63.6	50.2	41.32 (0.34)	40.3	19.3	medium
Portugal	1500	61.0	51.4	41.51 (0.35)	30.4	17.8	liberal
Romania	1500	31.0	50.0	41.11 (0.34)	56.9	27.1	medium
Spain	1645	50.3	49.8	40.90 (0.31)	66.8	12.2	liberal
Spain—Catalonia ^d^	661	51.1	49.6	41.70 (0.48)	72.3	12.6	liberal
Sweden	1623	35.9	50.9	40.76 (0.35)	56.9	34.8	strict
UK	1045	15.0	51.1	42.33 (0.58)	48.9	41.8	medium

^a^ unweighted sample size from the original survey; ^b^ “secondary education” combined general upper secondary education, vocational upper secondary education, post-secondary nontertiary education, and short-cycle tertiary education, “high education” combined Bachelor’s level, Master’s level, and Doctoral level of education; ^c^ national alcohol policy score derived from the AMPHORA project [21], ranking: “liberal” alcohol policy score ≤ 70 points, “medium” alcohol policy score 71 to 101 points, “strict” alcohol policy score 102 to 160 points; and ^d^ Spanish Autonomous Community of Catalonia. RARHA, Joint Action on Reducing Alcohol Related Harm; SEAS, Standardized European Alcohol Survey; UK, United Kingdom of Great Britain and Northern Ireland.

**Table 3 ijerph-16-04461-t003:** Results of the hierarchical regression models for gender, age, educational achievement, and drinking status as predictors for individual-level class endorsement probability (dependent variable) by class.

Variable.		Class 1		Class 2		Class 3	
		*OR*	95% CI	*OR*	95% CI	*OR*	95% CI
Gender (ref. male)							
	Female	1.47 ^‡^	[1.41; 1.52]	0.60 ^‡^	[0.58; 0.63]	1.26 ^‡^	[1.22; 1.32]
Age (ref. ≤34 years)							
	35–49 years	1.07 ^†^	[1.03; 1.12]	0.88 ^‡^	[0.84; 0.93]	1.09 ^‡^	[1.04; 1.15]
	≥50 years	1.04	[0.99; 1.09]	0.83 ^‡^	[0.78; 0.87]	1.25 ^‡^	[1.19; 1.31]
Educational achievement (ref. primary and lower secondary education)							
	Secondary education	1.07 *	[1.01; 1.13]	1.18 ^‡^	[1.11; 1.26]	0.77 ^‡^	[0.73; 0.81]
	High education	1.45 ^‡^	[1.36; 1.54]	1.12 ^†^	[1.04; 1.20]	0.53 ^‡^	[0.50; 0.57]
Drinking status (ref. abstainer)							
	Low-risk drinking	0.74 ^‡^	[0.70; 0.78]	2.25 ^‡^	[2.11; 2.41]	0.55 ^‡^	[0.53; 0.58]
	High-risk drinking	0.62 ^‡^	[0.55; 0.70]	2.33 ^‡^	[2.04; 2.67]	0.65 ^‡^	[0.58; 0.73]

Note: Class 1, “support of alcohol control policies”; class 2, “rejection of alcohol control policies”; and class 3, “acquiescence tendency”. Models were adjusted for national alcohol policy score and country identification number was used as random intercept. OR, odds ratio; CI, confidence interval; and ref., reference. * *p* < 0.05, ^†^
*p* < 0.01, and ^‡^
*p* < 0.001.

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
