# Peer review of "How Attitudes toward Alcohol Policies Differ across European Countries: Evidence from the Standardized European Alcohol Survey (SEAS)"

_ijerph, 2019, doi:10.3390/ijerph16224461_

Round 1
Reviewer 1 Report
An interesting manuscript in its field, I suggest that the authors add their next steps towards informing the public as they said they wanted to do at the end of their abstract.
suggest that the authors add the number of patients with depression and drug abuse among different countries. Moreover; the percentage of unemployment.
Author Response
Reviewer: An interesting manuscript in its field, I suggest that the authors add their next steps towards informing the public as they say they wanted to do at the end of their abstract
Answer: Thank you for your suggestion to describe next steps towards informing the public in the end of our article. In our manuscript, we suggest general measures to promote public attitudes towards alcohol control policies in the future. However, we do not recommend specific steps. This is because of two major reasons: Firstly, we suggest that promotion of public attitudes depend on the current status of implementing policies. We expect that different measures are required before a new policy will be implemented (e.g., acknowledgment and dissemination of evidence and likely impact by policy makers and journalist) compared to actions during the implementation process (e.g., monitoring of results). Secondly, such actions might be considerably different between European countries depending on the current level of policy endorsement (e.g., comparing Nordic countries with Eastern European countries) and cultural differences (e.g., wine drinking as part of the food culture in France).
We have addressed these arguments at the end of the article.
Reviewer: Suggest that the authors add the number of patients with depression and drug abuse among different countries. Moreover; the percentage of unemployment
Answer: Our data based on the Standardized European Alcohol Survey which only recorded questions on alcohol consumption, drinking behaviors, attitudes, policy endorsement, and sociodemographics. Unfortunately, no information are given for any depressive symptoms or (illegal) drug abuse. Therefore, we are not able to determine the number of patients among the different countries within our sample.
Regarding employment status, we were able to provide prevalence by location in Table 2. However, we chose to present the proportion of level of educational achievement (secondary education, high education) because we used educational achievement as indicator of socioeconomic status in the analyses.
Reviewer 2 Report
This as a well written paper considering across country and individual level differences in public attitudes to alcohol policy within Europe. The findings are interesting and provide an insight into public attitudes to alcohol policy and variation between different countries.
Only minor comments for improvement:
Abstract - include from the start that the paper is about attitudes to alcohol policy in the general public (it was not clear whose attitudes were being discussed)
Introduction: It would be good to include within the introduction the rationale for using a latent class analysis in contrast to previous work (on the same data?) which used factor analysis. The contrast between the findings with different methods is interesting and both provide a different insight into clustering of attitudes but the reason for deciding to do this is not given in the paper.
Author Response
Reviewer: Abstract - include from the start that the paper is about attitudes to alcohol policy in the general public (it was not clear whose attitudes were being discussed)
Answer: We have addressed this concern in the revised version and specified in the abstract that we focused on the public opinion towards alcohol policies.
Reviewer: Introduction: It would be good to include within the introduction the rationale for using a latent class analysis in contrast to previous work (on the same data?) which used factor analysis. The contrast between the findings with different methods is interesting and both provide a different insight into clustering of attitudes but the reason for deciding to do this is not given in the paper.
Answer: We have addressed this comment in the introduction of the article.
Reviewer 3 Report
Very nice study. Appropriate design and data analyses. I agree with the conclusions about educating the public about the effectiveness of these policies and the harm attributable to alcohol. I just completed a study that did just that and resulted in the majority of respondents supporting the strategy in a survey:
Fell, James (2019). Underutilized strategies in traffic safety: Results of a nationally representative survey. Traffic Injury Prevention, Special Issue: AAAM 2019.
Question:
With more and more people in most industrialized high income countries attaining higher education, does that mean more or less rejection of strict alcohol policies?
Page 4, line 118: Change "which" to 'with'
Page 9, line 266: Change "occurred" to "occur'
Author Response
Reviewer: With more and more people in most industrialized high-income countries attaining higher education, does that mean more or less rejection of strict alcohol policies?
Answer: We are not able to provide a response to this interesting question based on cross-sectional data analyses. However, the question goes along with our call to monitor the public opinion to detect changes in attitudes and possible influencing factors – such as changes in education. To answer your question, we included this as outlook for future research in the end of the article.
Reviewer: Page 4, line 118: Change “which” to ‘with’
Answer: Corrected in the revised version of the article.
Reviewer: Page 9, line 266: Change “occurred” to ‘occur’
Answer: Corrected in the revised version of the article.